# Disease burden and seasonal impact of improving rotavirus vaccine coverage in the United States: A modeling study

**Chin-En Ai** [1]*, **Molly Steele**[1], **Benjamin Lopman**[2]

**1** Department of Environmental Health, Rollins School of Public Health, Emory University, Atlanta, Georgia, United States of America, **2** Department of Epidemiology, Rollins School of Public Health, Emory University, Atlanta, Georgia, United States of America

* cai4@emory.edu

## Abstract

### Background

Prior to vaccine introduction in 2006, rotavirus was the leading cause of severe diarrhea in children under five years of age in the U.S. Vaccination of infants has led to major reductions in disease burden, a shift in the seasonal peak and the emergence of a biennial pattern of disease. However, rotavirus vaccine coverage has remained relatively low (70–75%) compared to other infant immunizations in the U.S. Part of the reason for this lower coverage is that children whose care is provided by family practitioners (FP) have considerably lower probability of being vaccinated compared to those seen be pediatricians (PE). We used a dynamic transmission model to assess the impact of improving rotavirus vaccine coverage by FP and/or PE on rotavirus gastroenteritis (RVGE) incidence and seasonal patterns.

### Methods

A deterministic age-structured dynamic model with susceptible, infectious, and recovered compartments (SIRS model) was used to simulate rotavirus transmission and vaccination. We estimated the reduction of RVGE cases by 2 doses of rotavirus vaccine with three vaccination scenarios: (Status Quo: 85% coverage by pediatricians and 45% coverage by family practitioners; Improved FP: 85% coverage by pediatricians and family practitioners; Improved FP+PE: 95% coverage by pediatricians and family practitioners). In addition, we tested the sensitivity of the model to the assumption of random mixing patterns between children visiting pediatricians and children visiting family practitioners.

### Results

In this model, higher vaccine coverage provided by family practitioners and pediatricians leads to lower incidence of severe RVGE cases (23% averted in Improved FP and 57% averted in Improved FP+PE compared to Status Quo) including indirect effects. One critical impact of higher total vaccine coverage is the effect on rotavirus epidemic patterns in the U. S.; the biennial rotavirus epidemic patterns shifted to reduced annual epidemic patterns.

**Data Availability Statement:** This is a simulation study, so there is no data aside from the model parameter inputs. All model code are available via https://www.protocols.io/view/disease-burden-

and-seasonal-impact-of-improving-ro-bba4iigw/abstract.

**Funding:** This work was supported by NIH/NIAID (R01-AI112970) and the Vaccine Impact Modeling Consortium. The funder had no role in study design, data collection and analysis, decision to publish, or preparation of the manuscript.

**Competing interests:** The authors have declared that no competing interests exist.

Additionally, assortative mixing patterns in children visiting pediatricians and family practitioners amplify the impact of increasing vaccine coverage.

## Conclusion

Other high-income countries that introduced vaccine have not experienced biennial patterns, like the U.S. Our results suggest that increasing overall vaccine coverage to 85% among infants would lead to an overall reduction in incidence with annual epidemic patterns.

## Introduction

Globally, rotavirus is the leading cause of severe diarrhea, hospitalization, and diarrhea related deaths in infants and children younger than 5 years old [1]. Before the introduction of rotavirus vaccines in 2006, rotavirus caused an estimated 200,000 emergency room visits, 55,000 to 70,000 hospitalizations, and 20 to 60 deaths annually in children younger than 5 years of age, leading to approximately $1 billion in direct and indirect costs to the U.S. [2].

Data from the National Respiratory and Enteric Virus Surveillance System (NREVSS) in the U.S. indicates that rotavirus gastroenteritis (RVGE) incidence has declined between 57% - 89% since the introduction of vaccines in 2006 [3]. Rotavirus hospitalization rates have reduced between 70% - 98%, and all cause diarrhea-associated hospitalization rates declined between 9% - 76% in children under the age of 5 [4]. Rotavirus vaccines have also provided indirect benefits to unvaccinated individuals across the age range [5].

By 2016, full two dose coverage of rotavirus vaccines reached 74.1% in U.S. children 19–35 months of age. However, rotavirus vaccine coverage is still lower than other routine childhood recommended vaccines (DTap≥3 doses: 95%, Poliovirus≥3 doses: 93.7%, MMR ≥1 doses: 91.1%, Heb≥3 doses: 90.5% in 2016) and below the Healthy People 2020 goal of 80% coverage [6, 7]. Therefore, to meet the Healthy People 2020 goal of 80% coverage [8] and decrease further rotavirus disease and economic burden, promotion of rotavirus vaccine coverage needs to be considered.

Prior to rotavirus vaccination, RVGE showed a winter-spring seasonality and geographic patterns that begin in the southwest in December-January, extending across to the U.S. and ending in the northeast during April-May. In the post-vaccine era, the rotavirus season was shorter, delayed and of smaller magnitude compared to seasons in the pre-vaccine era, but a biennial pattern of RVGE incidence emerged [3, 9, 10]. This RVGE reduction was accompanied by biennial patterns with higher seasonal peak in 2009, 2011, and 2013 compared to lower seasonal peak in 2008, 2010, and 2012 [3, 7]. In general, biennial patterns of RVGE could be induced by factors that govern the rate of new susceptibles. These include incomplete vaccine coverage rates, imperfect vaccine efficacy, and birth rates [11–13]. In one study, a RVGE transmission model predicted that biennial patterns of RVGE emerge after the introduction of vaccines when birth rates are low, while an annual pattern of RVGE was predicted when birth rates are high [12]. In the U.S, birth rates have remained fairly stable in the pre and post-vaccine era. Thus, birth rate is likely not a significant driver of the biennial pattern of RVGE in the U.S. Incomplete vaccination coverage that leads to a build-up of susceptible children may be driving this biennial pattern [13]. Unvaccinated children drive the higher rotavirus hospitalizations in the biennial patterns [5]. In contrast, other developed countries with high coverage of rotavirus vaccination (>85%), such as Belgium, Austria, Australia, Finland, and Germany, did not have the biennial epidemic patterns after the introduction of vaccines [14–18].

Health care providers play a critical role in promoting vaccines [19]. Pediatricians and family practitioners are two essential rotavirus vaccine providers in the U.S. Pediatricians provide rotavirus vaccine to all eligible infants at considerably higher levels than family practitioners do (85% and 45%, respectively) [20]. Family practitioners may be more concerned about vaccine safety and the burden of adding additional vaccines to the childhood vaccination schedule [20, 21]. Accordingly, family practitioners could play an essential role in increasing rotavirus vaccine coverage and could impact RVGE incidence in the U.S. [22].

Our research aim is to assess the potential impact of improved rotavirus vaccine coverage in terms of disease incidence and epidemiological patterns of rotavirus in children under 5 in the U.S. Our main hypothesis is that the incidence of RVGE in the U.S. can be further decreased, with a shift from biennial to annual cycles if family practitioners were to provide rotavirus vaccine at the same level of pediatricians. We adapted and analyzed a dynamic transmission model of rotavirus disease and vaccination that included heterogeneity in vaccine coverage and contact patterns among children provided care by pediatricians or family practitioners.

## Methods

### Model design and model parameters

We used a dynamic deterministic model of rotavirus transmission with susceptible, infectious, and recovered compartments (SIRS model) to represent rotavirus transmission and vaccination. This model is age-structured with six age groups: 0–1 month, 2–3 months, 4–11 months, 1–4 years, 5–24 years, and above 25 years. The model equations are shown in the S1 Appendix. Individuals are born with maternal immunity which wanes at rate $e$. Susceptible individuals are infected at a rate $\lambda$ and enter the infectious compartment. Infected individuals either recover from infection and gain long-term immunity at rate $\gamma$ or become susceptible to subsequent infections. Immunity wanes at rate $\omega$ and individuals become susceptible again. We also included seasonal forcing in our model.

The model assumes individuals can have up to four rotavirus infections with decreasing probabilities of infection, disease and severe disease ($\varepsilon_i$, $\alpha_i$, $\sigma_i$) for each subsequent exposure (Table 1) [23]. In addition, we assumed only symptomatic individuals are infectious and primary infections contribute more to transmission than subsequent infections [24–27]. Only primary and secondary rotavirus infection were assumed to develop severe diarrhea. Additionally, we assumed primary infection, secondary infection, tertiary infection, and quaternary infection have the same duration of infectiousness.

We assumed the proportion of children visiting pediatricians was 84.4% while the proportion of children visiting family practitioners was 15.6% based on data from the MarketScan Research Database [32]. We separated children under one year of age into pediatrician and family practitioner-attending groups in the model to predict the impact of different vaccine coverage in these two groups. In the model, children get the first dose vaccine at two months of age and second dose at four months of age. U.S. birth rates were informed by the CDC Wonder database [31]. We assumed maternal immunity does not have an effect on vaccine efficacy and assumed that death rates equaled the birth rate so that the total population remains constant. In baseline analysis, we assumed an assortative contact structure between different age groups based on the POLYMOD study [30].

### Parameters estimates

We estimated four age specific transmission parameters ($q_1$ for <1 year, $q_2$ for 1–4 years, $q_3$ for 5–24 years, and $q_4$ for > 25 years), seasonality parameters ($A, \theta$) and a reporting rate. We used

**Table 1. Natural history, demographic and estimated parameter values used in epidemiological model.**

| Parameter | Symbol | Parameter value | Description | Reference |
|---|---|---|---|---|
| Transmission probability | $q_i$ | $q_1 = 0.9998$<br>$q_2 = 0.4494$<br>$q_3 = 0.0472$<br>$q_4 = 0.0019$ | Probability of transmission per contact.<br>$q = 1...4$ represent age group <1 year, 1–4 years; 5–24 years, > 25 years, respectively | Estimated |
| Seasonal transmission amplitude | $A$ | 0.0866 | Proportion change in disease incidence | Estimated |
| Seasonal offset | $\theta$ | 0.4942 | | Estimated |
| Reporting rate | $\delta$ | 0.0538 | Probability that severe RVGE case is reported | Estimated |
| Vaccine Efficacy | $\psi$ | 0.5 | | Calibrated |
| Daily rate of loss of immunity | $\omega$ | 1/21,154 | Rate at which immune individuals become re-susceptible infection | Atchison, 2010 [28] |
| Daily rate of loss of maternal immunity | $e$ | 1/90 | Maternal immunity against rotavirus infection wane at a constant rate on average 90 days | Heymann, 2015 [29] |
| Daily rate of loss of infection | $\gamma$ | 1/5 | Symptoms last 2–7 days but on average 5 days | Heymann, 2015 [29] |
| Risk of infection after previous infection | $\varepsilon_i$ | $\varepsilon_1 = 0.62$<br>$\varepsilon_2 = 0.37$<br>$\varepsilon_3 = 0.37$ | After first infection<br>After second infection<br>After third infection | Velazquez et al., 1996 [23] |
| Proportion of symptomatic infection in nth infection | $\alpha_i$ | $\alpha_1 = 0.47$<br>$\alpha_2 = 0.25$<br>$\alpha_3 = 0.32$<br>$\alpha_4 = 0.20$ | At first infection<br>At second infection<br>At third infection<br>At fourth infection | Velazquez et al., 1996 [23] |
| Proportion of symptomatic infection associated with severe disease at nth infection | $\sigma_i$ | $\sigma_1 = 0.28$<br>$\sigma_2 = 0.19$ | At first infection<br>At second infection | Velazquez et al., 1996 [23] |
| Relative infectiousness of non-primary infections | $r$ | $r = 0.25$ | | Velazquez et al., 1996 [23] |
| Daily aging rates for age group j | $a_j$ | $a_1 = 1/60$<br>$a_2 = 1/120$<br>$a_3 = 1/365$<br>$a_4 = 1/7,300$ | $j = 1...4$ represent age group 0–3 months, 4–11 months, 1–4 years, and 5–24 years, respectively | |
| Counts of total contacts | $c_i$ | $c_1 = 5.43$<br>$c_2 = 8.56$<br>$c_3 = 15.65$<br>$c_4 = 14.16$ | Counts for age <1 year<br>Counts for age 1–4 years<br>Counts for age 5–24 years<br>Counts for age >25 years | Mossong et al., 2008 [30] |
| Birth rate (Daily) | $\mu$ | 1/30,827.7 | U.S. 2017 birth rate | CDC Wonder [31] |

maximum likelihood by fitting the model to data on monthly counts of severe RVGE cases from the MarketScan Research Database. All analyses were conducted using the statistical program R version 1.1.423 [33], and the *deSolve* package to solve differential equations [34]. We calibrated a parameter for rotavirus vaccine efficacy to allow the model to capture observed biennial epidemic patterns (Table 1). Biennial epidemic patterns in the U.S. may be driven by a slower accumulation of susceptibles following modest vaccine coverage. By calibrating the vaccine efficacy and rate of loss of immunity parameters (i.e., increasing vaccine efficacy and lowering the rate of loss of immunity) we can slow the accumulation of the simulated susceptible population which allows the model to capture biennial epidemic patterns.

## Vaccine scenario

We estimated the impact on severe RVGE cases of three vaccination scenarios, all based on 2-dose coverage: 85% coverage of the pediatrician (PE) population and 45% coverage of the family practitioner (FP) population, which is the present vaccine coverage (Status Quo); 85% coverage of the PE population and 85% coverage of the FP population, (Improved FP

Coverage); and 95% coverage of the PE population and FP populations, (Improved FP + PE). For the Improved FP and Improved FP + PE scenarios, we assumed a change in vaccine coverage beginning in 2018. As outcomes, we calculated the percent of severe RVGE cases averted by comparing the rate of severe RVGE cases in Improved FP and Improved FP + PE to the average rate of severe RVGE cases in 2000–2006, prior to the introduction of vaccines, and each year from 2007 to 2030 in Status Quo. To quantify the impact of higher vaccine coverage on severe RVGE on the Status Quo population, the percent of severe RVGE cases averted for each vaccination scenario compared to Status Quo was calculated as follows:

$$Percent\ averted = 100\% * \left( \frac{Rate\ of\ severe\ case_{Status\ Quo} - Rate\ of\ severe\ case_i}{Rate\ of\ severe\ case_{Status\ Quo}} \right)$$

where $i$ is the vaccine scenario ($i$ = 2, 3). To quantify the impact of higher vaccine coverage in the pre-vaccine era, the percent of severe RVGE cases averted for each rotavirus vaccination scenario compared to pre-vaccine era was calculated as follows:

$$Percent\ averted = 100\% * \frac{Rate\ of\ severe\ case_{pre-vaccine} - Rate\ of\ severe\ case_i}{Rate\ of\ severe\ case_{pre-vaccine}}$$

where $i$ is the vaccine scenario ($i$ = 1, 2, 3).

### Sensitivity analysis to assumptions about mixing patterns

Initially we assumed random mixing patterns between children visiting PEs and FPs. However, it is possible that there is assortative mixing within these groups such that children who attend FPs are more likely to mix with other such children. Therefore, we tested the sensitivity of the model to this assumption of mixing by setting contact within a group to be higher than contact between groups to depict an assortative mixing pattern. We assumed that 80% of contacts occur within a group and 20% of contacts occur between groups in assortative mixing patterns.

## Results

### Incidence reduction of severe RVGE

Before rotavirus vaccine introduction, the model estimated that the average incidence of severe RVGE cases was 327 per 10,000 population between 2000 and 2006 (S1 Table in S1 Appendix). To account for biennial patterns of incidence post vaccine introduction, we present four-year averages of severe RVGE cases to compare the incidence of severe RVGE cases in new vaccine scenarios to Status Quo. The four-year average rates of severe RVGE cases in Status Quo, Improved FP, and Improved FP+PE were 75, 57, and 33 per 10,000 population, respectively (Table 2). The four-year average percent of severe RVGE averted for Improved FP and Improved FP+PE compared to Status Quo were 23% and 57%, respectively.

In Status Quo, biennial patterns were observed with incidence at around 80 and 65 cases per 10,000 in odd and even years, respectively; The model predicted that rotavirus epidemic patterns shift from biennial epidemic patterns to reduced annual epidemic in Improved FP and Improved FP+PE (Fig 1).

### Indirect benefits of improved rotavirus vaccine coverage in population attending family practitioners

In children 0–11 months old and Improved FP, the model estimated the 2018–2029 average percent of severe RVGE cases averted were 23% in the PE population and 37% in the FP population compared to Status Quo (S2 Table in S1 Appendix). The percent of severe RVGE cases

averted in the PE population compared to the pre-vaccine era was 85% in Status Quo and 89% in Improved FP. Since there is no improved vaccine coverage in the PE population in Improved FP, this additional 4% of severe cases (about 7000 cases) averted in PE population are indirect benefits of improved vaccine coverage from FP population.

### Sensitivity to assumptions about mixing patterns of children attending pediatricians and family practitioners

In the sensitivity analysis (S3 Table in S1 Appendix) we assumed assortative mixing patterns with 80% of contacts occurring within a group and 20% of contacts occurring between groups. This resulted in an increase of 5 per 10,000 population of the four-year average severe RVGE incidence (Table 3) compared to random mixing patterns. However, assuming assortative mixing resulted in around 5% greater reduction in the four-year average of severe RVGE cases averted in Improved FP and Improved FP + PE, respectively, compared to Status Quo than the random mixing patterns (Table 3). The epidemic patterns of RVGE stayed the same as random mixing patterns (Fig 2).

## Discussion

We modeled the epidemiological impact of higher vaccine coverage by U.S. family practitioners and pediatricians and note several key findings. First, as expected, higher coverage leads to lower incidence of severe RVGE cases through both direct and indirect vaccine effects. Second, under both Improved FP and Improved FP + PE vaccine coverage scenarios, we predicted current biennial rotavirus epidemic patterns would shift to more predictable, annual epidemic patterns. Third, while the mixing patterns of populations attending FPs versus PEs is unknown, we found that assuming assortative mixing among children visiting PEs and FPs amplified the impact of increasing vaccine coverage.

A number of results from this model are consistent with what has been predicted with previously published rotavirus transmission models; high rotavirus vaccine coverage (>85%) resulted in reductions of annual severe RVGE incidence by 56% [35], 70% [28, 36] and 84% [37] for children under 5 years of age, compared to pre-vaccination levels in different model studies. One modeling study for Germany [37] predicted no rotavirus biennial epidemic patterns even after high national rotavirus vaccine coverage (90%). However, other models had different predicted effects on rotavirus epidemic patterns after high national rotavirus vaccine

**Table 2. Four-year average incidence rates and percent of severe RVGE cases averted in new vaccination strategies assuming random mixing patterns between children visiting pediatricians and family practitioners.**

| Time period | Vaccine Scenario | | |
|---|---|---|---|
| | Status Quo [a] Rate[d] | Improved FP [b] Rate[d] (Averted)[e] | Improved FP + PE[c] Rate[d] (Averted)[e] |
| 2018–2021 | 73 | 57 (23%) | 35 (52%) |
| 2022–2025 | 74 | 56 (24%) | 32 (57%) |
| 2026–2029 | 74 | 57 (23%) | 33 (56%) |

a. 85% vaccination coverage for children visiting PEs and 45% for children visiting FPs (total 78.6% current vaccination coverage).

b. 85% vaccination coverage for children visiting PEs and FPs.

c. 95%vacccination coverage for children visiting PEs and FPs.

d. Rate of severe RVGE per 10,000 population

e. Percent of severe RVGE averted compared to Status Quo in same time period

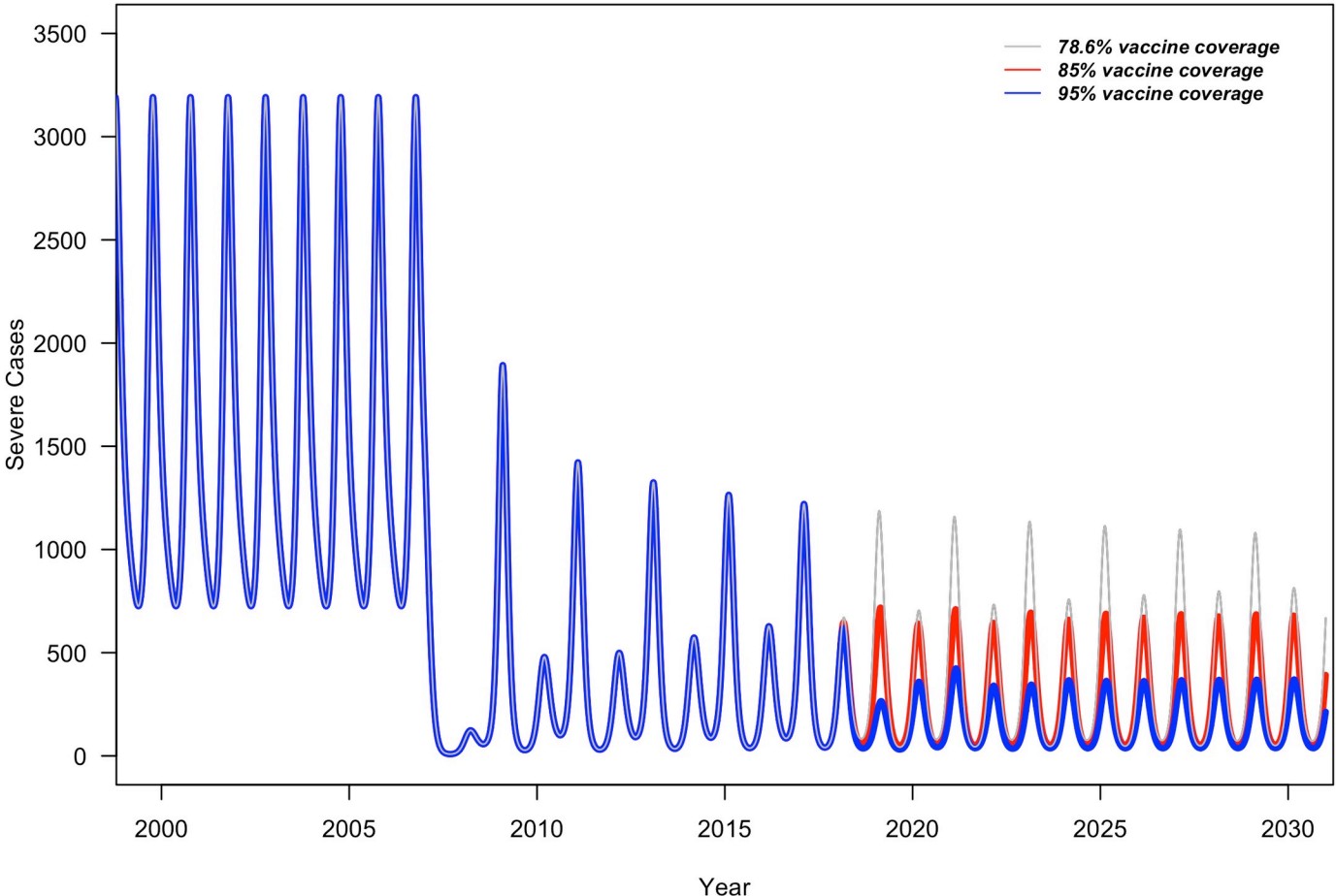

**Fig 1.** Monthly number of severe RVGE cases in children under 5 years of age with Status Quo (grey), Improved FP (red), and Improved FP + PE (blue) vaccine coverage assuming random mixing patterns between children visiting PEs and FPs.

coverage (90%) [28, 36, 38]. These models predicted biennial epidemic patterns in medium vaccine coverage (70%) and elimination of rotavirus at 90% coverage, whereas other models predicted potential biennial epidemic patterns with 90% vaccine coverage. Differences in the

**Table 3. Four-year average incidence rates and percent of severe RVGE cases averted in new vaccination strategies with assortative mixing patterns assuming 80% of contacts occur within a group and 20% of contacts occur between groups.**

| Time period | Status Quo [a] Rate[d] | Improved FP [b] Rate[d] (Averted)[e] | Improved FP + PE[c] Rate[d] (Averted)[e] |
|---|---|---|---|
| 2018–2021 | 78 | 55 (30%) | 34 (56%) |
| 2022–2025 | 79 | 57 (28%) | 31 (60%) |
| 2026–2029 | 79 | 57 (28%) | 33 (58%) |

a. 85% vaccination coverage for children visiting PEs and 45% for children visiting FPs (total 78.6% current vaccination coverage).

b. 85% vaccination coverage for children visiting PEs and FPs.

c. 95%vacccination coverage for children visiting PEs and FPs.

d. Rate of severe RVGE per 10,000 population

e. Percent of severe RVGE averted compared to Status Quo in same time period

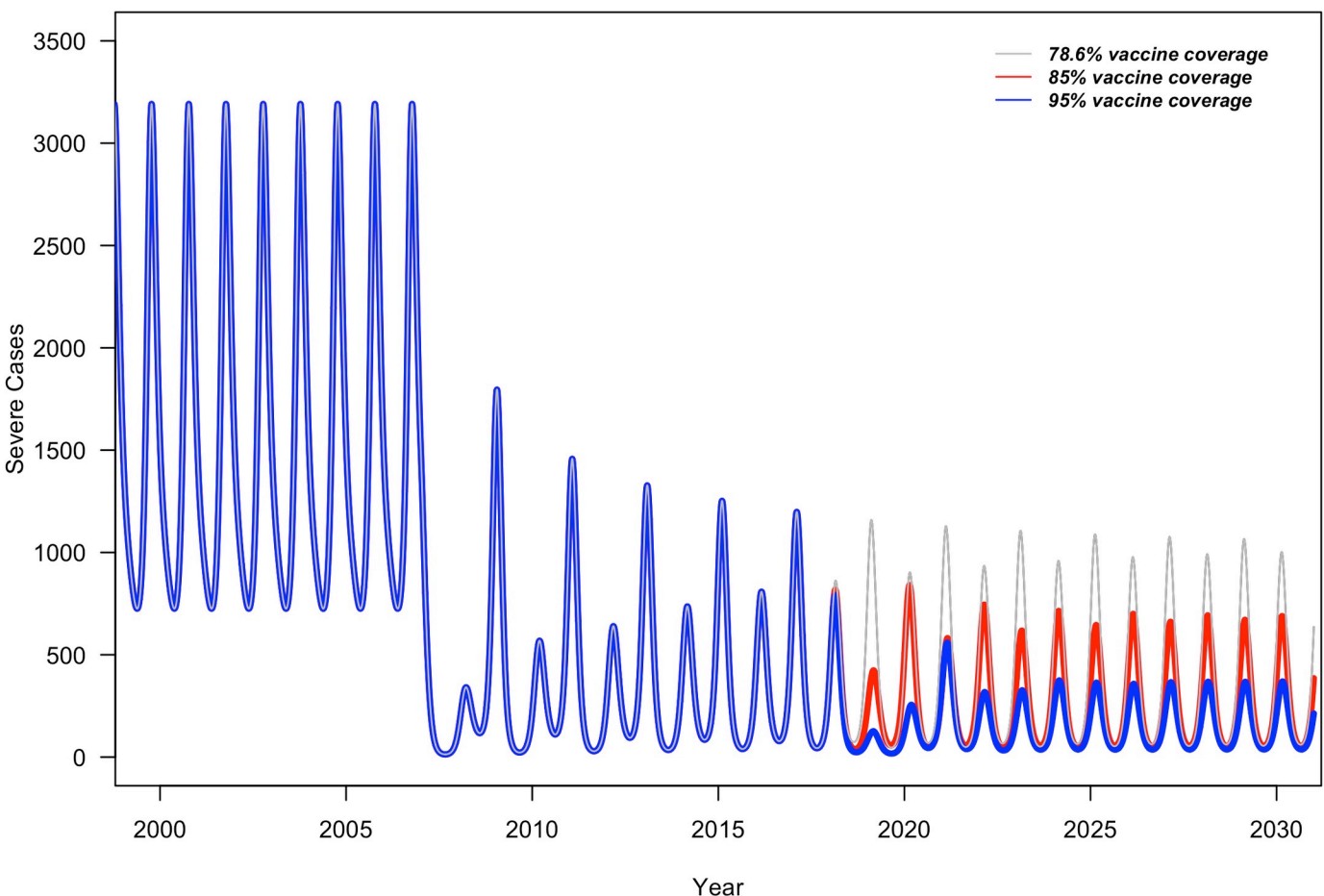

**Fig 2.** Monthly number of severe RVGE cases in children under 5 years of age with Status Quo (grey), Improved FP (red), and Improved FP + PE (blue) vaccine coverage with assortative mixing patterns assuming 80% of contacts occur within a group and 20% of contacts occur between groups.

assumptions and parameters of these models may explain the different predictions for epidemic patterns. For example, in our model we assumed vaccines and natural infection induce partial immunity (i.e. subsequent infection can still occur but at reduced rate) and a long duration of immunity. Another modeling study that predicted high vaccine coverage with biennial patterns assumed one-year duration of complete immunity after previous infection, then, individuals were assumed to be susceptible to infection again with partial immunity [38]. Moreover, our model fit to data from the U.S. whereas other rotavirus models were fit to rotavirus data from England and Wales. Differences in demographic conditions between the U.S. and England and Wales may also partly explain the differences in these results.

The emergence of biennial epidemic patterns of RVGE incidence in the U.S. after the introduction of a national vaccine program may be driven by accumulation of susceptible following modest vaccine coverage. Other developed countries with high coverage of rotavirus vaccination (>85%), such as Belgium, Austria, Australia, Finland, and Germany, have not experienced the biennial epidemic patterns after vaccine introduction [14–18]. Our model predicted that RVGE incidence shifted from a biennial pattern to an annual pattern when vaccine coverage reached 85%. Thus, promoting rotavirus vaccination in children visiting FPs lowers disease incidence rates and results in a shift from biennial to annual epidemic patterns. Higher vaccination coverage leads to a smaller susceptible population compared to the susceptible

population in Status Quo. Therefore, the rate of accumulation of susceptible children over two successive birth cohorts may not be sufficient to drive biennial 'mini-epidemics'. Shah et al. suggested that increased rotavirus vaccine coverage in the U.S. may change rotavirus epidemiological patterns since the biennial epidemic patterns may be driven by lower vaccine coverage [13]. The shift of epidemic patterns from biennial patterns to reduced annual patterns may be beneficial for public health preparedness as the burden of disease would be more consistent year to year.

Our predictions were somewhat sensitive to assumptions about mixing patterns in children visiting PEs and FPs. When we assumed assortative mixing, we found greater impact of increasing vaccine coverage on severe RVGE incidence. Our model had a higher force of infection with assortative mixing patterns than that in random mixing patterns. Effelterre et al. found that the estimated basic reproductive number ($R_0$) of rotavirus is higher when mixing patterns are assumed to be assortative. However, that study showed that reduction in any grade of severe RVGE incidence in children under 5 years of age after vaccination is higher when the assortative mixing is lower, which contradicts our results [39]. Effelterre et al. focused on the reduction of any RVGE but our study focused on the reduction of severe RVGE. This difference influences the vaccine impact on the reduction of RVGE since rotavirus vaccines have better efficacy against severe RVGE [4, 40, 41]. In another study, Choe and Lee indicated that the higher degree of assortative mixing had higher $R_0$ than random mixing [42]. Furthermore, after vaccine introduction, a higher degree of assortative mixing resulted in lower incidence over time. We have no data to inform mixing patterns between PEs and PFs populations. But, under either scenario, promoting vaccine coverage in children visiting FPs can have significant impacts on reducing severe RVGE incidence in children under 5 years of age in the U.S. Future rotavirus vaccine promotion strategies in the U.S. can emphasize covering children visiting FPs.

Our model had several limitations. First, though we took assortative contact patterns for children visiting PEs and FPs into account, there currently are no data that describe the true assortative contact patterns within and between patients who attend physician groups. Furthermore, the assortative contact structure between different age groups used in this model is based on the POLYMOD study, which is a population-based contact survey in Europe. This may not accurately represent the contact structure in the U.S. Second, while our model did capture the intensity in the year to year variability in the present biennial epidemic patterns, the observed RVGE cases in even rotavirus season years were much lower than the value predicted by our model [4, 43]. Third, we fit this model to a large, commercial insurance dataset. While this database covers most states, it may not be representative of the whole U.S. population. For example, children who fall under the coverage of Medicaid, may have lower childhood vaccination coverage and higher incidence of RVGE than the children represented in the MarketScan commercial insurance database [6].

In conclusion, we used a dynamic transmission model to predict the impact of increasing immunization rates among children who attend family practitioners. Under these higher vaccine coverage levels, we predicted that biennial patterns would shift to annual patterns with lower magnitude of RVGE incidence.

## Supporting information

**S1 Appendix.** S1 Table. Rate of severe rotavirus cases (per 10,000) and percent of severe cases averted for each rotavirus vaccination scenario in 2000 to 2030 assuming random mixing patterns between children visiting pediatricians and family practitioners. S2 Table. Children 0–11 months old percent of severe rotavirus cases averted in post-vaccine era and new vaccine

scenario after 2018 in pediatrician and family practitioner populations assuming random mixing patterns between children visiting pediatricians and family practitioners. S3 Table. Rate of severe rotavirus cases (per 10,000) and percent of severe cases averted for each rotavirus vaccination scenario in 2000 to 2030 with assortative mixing patterns assuming 80% of contacts occur within a group and 20% of contacts occur between groups.
(DOCX)

## Author Contributions

**Conceptualization:** Chin-En Ai, Molly Steele, Benjamin Lopman.

**Data curation:** Molly Steele.

**Formal analysis:** Chin-En Ai.

**Investigation:** Chin-En Ai, Molly Steele.

**Methodology:** Chin-En Ai, Molly Steele.

**Project administration:** Molly Steele, Benjamin Lopman.

**Software:** Chin-En Ai.

**Supervision:** Benjamin Lopman.

**Validation:** Chin-En Ai.

**Visualization:** Chin-En Ai.

**Writing – original draft:** Chin-En Ai.

**Writing – review & editing:** Chin-En Ai, Molly Steele, Benjamin Lopman.

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
