## [Decision Letter · Decision Letter 0]

2 Dec 2019

PONE-D-19-26596

Disease burden and seasonal impact of improving rotavirus vaccine coverage in the United States: a modeling study

PLOS ONE

Dear Mr. Ai,

Thank you for submitting your manuscript to PLOS ONE. After careful consideration, we feel that it has merit but does not fully meet PLOS ONE’s publication criteria as it currently stands. Therefore, we invite you to submit a revised version of the manuscript that addresses the points raised during the review process.

We would appreciate receiving your revised manuscript by Jan 16 2020 11:59PM. To enhance the reproducibility of your results, we recommend that if applicable you deposit your laboratory protocols in protocols.io, where a protocol can be assigned its own identifier (DOI) such that it can be cited independently in the future. For instructions see: http://journals.plos.org/plosone/s/submission-guidelines#loc-laboratory-protocols

We look forward to receiving your revised manuscript.

Kind regards,

Constantinos I. Siettos, Ph.D.

Academic Editor

PLOS ONE

Journal Requirements:

'This work was supported by NIH/NIAID (R01-AI112970) and the Vaccine Impact Modeling Consortium.'

'NO - Include this sentence at the end of your statement: The funders had no role in study design, data collection and analysis, decision to publish, or preparation of the manuscript.'

Please amend your Financial disclosure statement to declare sources of funding, or state that the authors received no specific funding.

Please provide an amended Funding Statement that declares *all* the funding or sources of support received during this specific study (whether external or internal to your organization) as detailed online in our guide for authors at http://journals.plos.org/plosone/s/submit-now.  

Please state what role the funders took in the study.  If any authors received a salary from any of your funders, please state which authors and which funder. If the funders had no role, please state: "The funders had no role in study design, data collection and analysis, decision to publish, or preparation of the manuscript."

Reviewers' comments:

Reviewer's Responses to Questions

**Comments to the Author**

1. Is the manuscript technically sound, and do the data support the conclusions?

Reviewer #1: Yes

Reviewer #2: Yes

2. Has the statistical analysis been performed appropriately and rigorously? 

Reviewer #1: N/A

Reviewer #2: Yes

3. Have the authors made all data underlying the findings in their manuscript fully available?

Reviewer #1: No

Reviewer #2: Yes

4. Is the manuscript presented in an intelligible fashion and written in standard English?

Reviewer #1: Yes

Reviewer #2: Yes

5. Review Comments to the Author

Reviewer #1: In this paper, Ai et al use a standard mathematical model to investigate the impact of improving coverage with rotavirus vaccine on rotavirus disease burden, and the effect it could have on the biennial patterns that have emerged post vaccine introduction. They test increasing coverage in two patient populations: those attended by pediatricians and those attended by family practitioners. They project significant reductions in disease burden (compared to current levels) when coverage is increased among patients of family practitioners alone, and larger reductions when coverage is increased among patients of both family practitioners and pediatricians. With increased coverage, they note a disappearance of the biennial pattern of rotavirus incidence (it is replaced by a reduced annual pattern), supporting the hypothesis that biennial patterns arise from an inter-year accumulation of susceptible persons in settings with moderate coverage. This is a well written paper, that is concise and understandable even for non-modelers, with clear conclusions that are directly derived from their results, and with an important public health message. The model structure has been built to emulate natural rotavirus disease and transmission. I only have a few minor comments.

Introduction:

Line 60: Typographical error, it should say “$1 billion in direct and indirect costs to the U.S.”

Line 63: Typographical error, it should say “rotavirus incidence has declined between 57%” (erase ‘in the’)

Methods:

Line 110 and Table 1: The text mentions the model is age-structured with 6 age-groups, yet the table shows parameters only for 4 age-groups, would clarify. Perhaps estimated transmission parameters are reported as larger age-groups.

Line 145: Would expand on, explain more, how the calibration of vaccine effectiveness allows the model to capture the biennial patterns.

Discussion:

Line 278: The line “partial immunity when individuals are susceptible” might need to be rephrased, this is not clear.

Line 311: The public health relevance of the additional 4 percent in reductions in severe rotavirus gastroenteritis when assuming assortative transmission by practitioner is unknown, and authors might want to acknowledge this, or translate to absolute numbers.

Line 320: If the intensity in the year to year variability of the biennial patterns were not captured, then the authors might want to acknowledge that other factors might be at play, or hypothesize which ones.

Reviewer #2: The authors focus on an important aspect of mechanisms which may influence the effectiveness of vaccines in general and in the particular setting of RV vaccination. To estimate the reduction of RVGE incidence by efforts to improve vaccination rates by FP by applying a deterministic age-structured dynamic model is a feasible approach. Overall, this is a well-written manuscript with sound results.

However, I have some comments to deal with:

I doubt whether the biennial pattern of RVGE in U.S. may be explained only by incomplete vaccine coverage. Authors should provide more evidence for this hypothesis by literature or own experience. Trend for biennal patterns or oscillations in RVGE associated hospitalizations was seen by others (e.g. Prelog M et al., J Inf Dis 2016). Authors should comment on this. The immunological or virological mechanisms behind biennial patterns or disappearance of biennial patterns should be explained to understand the association with vaccine coverage.

Assumptions in table 1 are mainly based on few articles. Authors should comment why they took this choice and did not perform a systematic review approach previous to the modeling procedure. Authors should state why the articles by Velazquez or Heymann are prior to others if there are any on these parameters.

6. PLOS authors have the option to publish the peer review history of their article (what does this mean?). If published, this will include your full peer review and any attached files.

Reviewer #1: No

Reviewer #2: No

---

## [Author Response · Author response to Decision Letter 0]

15 Jan 2020

Response to Reviewers

Comments from the reviewers:

Reviewer #1: In this paper, Ai et al use a standard mathematical model to investigate the impact of improving coverage with rotavirus vaccine on rotavirus disease burden, and the effect it could have on the biennial patterns that have emerged post vaccine introduction. They test increasing coverage in two patient populations: those attended by pediatricians and those attended by family practitioners. They project significant reductions in disease burden (compared to current levels) when coverage is increased among patients of family practitioners alone, and larger reductions when coverage is increased among patients of both family practitioners and pediatricians. With increased coverage, they note a disappearance of the biennial pattern of rotavirus incidence (it is replaced by a reduced annual pattern), supporting the hypothesis that biennial patterns arise from an inter-year accumulation of susceptible persons in settings with moderate coverage. This is a well written paper, that is concise and understandable even for non-modelers, with clear conclusions that are directly derived from their results, and with an important public health message. The model structure has been built to emulate natural rotavirus disease and transmission. I only have a few minor comments.

Response: Thank you for your comments.

Introduction:

Line 60: Typographical error, it should say “$1 billion in direct and indirect costs to the U.S.”

Response: The typographical error is corrected 

Edits to text (line 60): “$1 billion in direct and indirect costs to the U.S.”

Line 63: Typographical error, it should say “rotavirus incidence has declined between 57%” (erase ‘in the’)

Response: The typographical error is corrected 

Edits to text (line 63): ” that rotavirus gastroenteritis (RVGE) incidence has declined between 57% - 89%”

Methods:

Line 110 and Table 1: The text mentions the model is age-structured with 6 age-groups, yet the table shows parameters only for 4 age-groups, would clarify. Perhaps estimated transmission parameters are reported as larger age-groups.

Response: Indeed, age groups 0 – 1 months, 2 – 3 months, and 4 – 11 months in the model shared the same group of estimated age specific transmission parameters: age-group <1 year. 

Edits to text (line 143 – 144):” We estimated four age specific transmission parameters (q_1 for <1 year, q_2 for 1-4 years, q_3 for 5-24 years, and q_4 for > 25 years)”

Line 145: Would expand on, explain more, how the calibration of vaccine effectiveness allows the model to capture the biennial patterns.

Response: A likely cause of biennial patterns of disease in the U.S. is the accumulation of susceptibles that results from modest vaccine coverage. Vaccine efficacy and the rate of loss of immunity both impact the accumulation of susceptibles over time so calibrating these parameters allows the model to capture the biennial patterns that are observed in the U.S. 

We add the explanation of the factors driving biennial patterns to the introduction and methods to explain why we calibrate vaccine effectiveness and rate of loss immunity to capture biennial patterns. Edits to text (lines 83 – 85): “In general, biennial patterns of RVGE could be induced by factors that govern the rate of new susceptibles. These include incomplete vaccine coverage, imperfect vaccine efficacy, and birth rate (Pitzer 2009; Pitzer 2011, Shah 2018). 

Edits to text (156 – 164): “Biennial epidemic patterns in the U.S. may be driven by a slower accumulation of susceptibles following modest vaccine coverage. By calibrating the vaccine efficacy and rate of loss of immunity parameters (i.e., increasing vaccine efficacy and lowering the rate of loss of immunity) we can slow the accumulation of the simulated susceptible population which allows the model to capture biennial epidemic patterns.”

Discussion:

Line 278: The line “partial immunity when individuals are susceptible” might need to be rephrased, this is not clear.

Response: In lines 286 – 288, partial immunity (i.e. subsequent infection can still occur but at reduced rate) was explained

Edits to text (lines 288 – 291): Another modeling study that predicted high vaccine coverage with biennial patterns assumed one-year duration of complete immunity after previous infection, then, individuals are susceptible to infection again with partial immunity

Line 311: The public health relevance of the additional 4 percent in reductions in severe rotavirus gastroenteritis when assuming assortative transmission by practitioner is unknown, and authors might want to acknowledge this, or translate to absolute numbers.

Response: Our raw data shows that an additional 7000 RVGE cases are averted when assuming assortative mixing patterns compared to random mixing patterns in Improved FP. 

Edits to text (lines 233 -235): Since there is no improved vaccine coverage in the PE population in Improved FP, this additional 4% of severe cases (7000 cases lower) averted in PE population are indirect benefits of improved vaccine coverage from FP population. 

Line 320: If the intensity in the year to year variability of the biennial patterns were not captured, then the authors might want to acknowledge that other factors might be at play, or hypothesize which ones.

Response: We rephrased our statement: Our model did capture the intensity in the year to year variability in the present biennial epidemic patterns, however, the actual RVGE cases in even year rotavirus season were much lower than the value predicted by our model (Aliabadi 2015; Getachew 2018). 

Edits to text (lines 333 – 336): Second, relatedly, our model did capture the intensity in the year to year variability in the present biennial epidemic patterns, however, the actual RVGE cases in even year rotavirus season were much lower than the value predicted by our model.

Reviewer #2: The authors focus on an important aspect of mechanisms which may influence the effectiveness of vaccines in general and in the particular setting of RV vaccination. To estimate the reduction of RVGE incidence by efforts to improve vaccination rates by FP by applying a deterministic age-structured dynamic model is a feasible approach. Overall, this is a well-written manuscript with sound results.

However, I have some comments to deal with:

I doubt whether the biennial pattern of RVGE in U.S. may be explained only by incomplete vaccine coverage. Authors should provide more evidence for this hypothesis by literature or own experience. Trend for biennal patterns or oscillations in RVGE associated hospitalizations was seen by others (e.g. Prelog M et al., J Inf Dis 2016). Authors should comment on this. The immunological or virological mechanisms behind biennial patterns or disappearance of biennial patterns should be explained to understand the association with vaccine coverage.

Assumptions in table 1 are mainly based on few articles. Authors should comment why they took this choice and did not perform a systematic review approach previous to the modeling procedure. Authors should state why the articles by Velazquez or Heymann are prior to others if there are any on these parameters.

Response: Thank you for your comments and suggestions 

Response to biennial patterns:

In general, biennial patterns of RVGE could be caused by(a combination of) vaccine coverage, vaccine efficacy, and birth rate (Pitzer 2009; Pitzer 2011). Since birth rates in the U.S have been low and fairly stable through the pre and post – vaccine era, and previous studies have indicated that incomplete vaccine coverage can drive biennial patterns, we considered vaccine coverage as the main factor in our model driving biennial patterns in the U.S. Note that these patterns do not occur in most other high-income countries with rotavirus immunization. We incorporated a detailed explanation of this rationale in the text (lines 83-89). Additionally, we also mentioned the disappearance of biennial patterns due to increasing of vaccine coverage predicted in the Weidemann 2014 modeling study in the text (lines 288 – 289)

Edits to text (lines 83 – 89): 

In general, biennial patterns of disease could be caused by a combination of vaccine coverage rates, vaccine efficacy, and birth rate, such that susceptibles are accumulated at a certain rate (Pitzer 2009; Pitzer 2011, Shah 2018). In one study, a RVGE transmission model predicted that biennial patterns of RVGE after introduction of vaccine when birth rates are lower while an annual pattern of RVGE was predicted when birth rates are high (Pitzer 2011). In the U.S, birth rates have remained fairly stable0 in the pre and post-vaccine era. Thus, birth rate is likely not a significant driver of the biennial pattern of RVGE in the U.S. 

Response to parameters selection:

We used data from the Velazquez 1996 study to inform the proportions of infection, symptomatic infection and severe disease and relative infectiousness of non-primary infections because of its rigorous study design and study population. Velazquez 1996 was a birth cohort study conducted in Mexico quantifying risk reduction after infection by collecting stool weekly from 200 infants from birth to 2-year-old. There are three birth cohort studies conducted in Mexican, Indian and African birth cohorts that quantify the reduced risk of subsequent infections. As there are no published birth cohort studies conducted in the U.S., we selected the Velazquez study of the Mexican birth cohort which would have more similar demographic characteristics to the U.S. than the birth cohorts from India and Africa. Moreover, the data from the Velazquez study has been used to parameterize many other rotavirus transmission models (Pitzer 2012; Atchison 2010; Atkins 2012).

The work by Prelog et al (Jid 2018) is an important paper showing the range of indirect effects of rotavirus immunization in Australia, but does not provide parameter input for our model. 

The Hymann 2015 study provides data on the natural history of rotavirus such as daily rate of loss of maternal immunity and daily rate of loss infection. These data have been used to inform parameters in several transmission model (Atchison 2010, Atkins 2012).

---

## [Decision Letter · Decision Letter 1]

28 Jan 2020

Disease burden and seasonal impact of improving rotavirus vaccine coverage in the United States: a modeling study

PONE-D-19-26596R1

Dear Dr. Ai,

We are pleased to inform you that your manuscript has been judged scientifically suitable for publication and will be formally accepted for publication once it complies with all outstanding technical requirements.

With kind regards,

Constantinos I. Siettos, Ph.D.

Academic Editor

PLOS ONE

Additional Editor Comments (optional):

Reviewers' comments:

Reviewer's Responses to Questions

**Comments to the Author**

1. If the authors have adequately addressed your comments raised in a previous round of review and you feel that this manuscript is now acceptable for publication, you may indicate that here to bypass the “Comments to the Author” section, enter your conflict of interest statement in the “Confidential to Editor” section, and submit your "Accept" recommendation.

Reviewer #1: All comments have been addressed

Reviewer #2: All comments have been addressed

2. Is the manuscript technically sound, and do the data support the conclusions?

Reviewer #1: Yes

Reviewer #2: Yes

3. Has the statistical analysis been performed appropriately and rigorously? 

Reviewer #1: N/A

Reviewer #2: Yes

4. Have the authors made all data underlying the findings in their manuscript fully available?

Reviewer #1: Yes

Reviewer #2: Yes

5. Is the manuscript presented in an intelligible fashion and written in standard English?

Reviewer #1: Yes

Reviewer #2: Yes

6. Review Comments to the Author

Reviewer #1: (No Response)

Reviewer #2: All comments have been addressed and there are no further questions or concerns regarding the manuscript.

7. PLOS authors have the option to publish the peer review history of their article (what does this mean?). If published, this will include your full peer review and any attached files.

Reviewer #1: No

Reviewer #2: No

---

## [Editor Report · Acceptance letter]

6 Feb 2020

PONE-D-19-26596R1 

Disease burden and seasonal impact of improving rotavirus vaccine coverage in the United States: a modeling study 

Dear Dr. Ai:

I am pleased to inform you that your manuscript has been deemed suitable for publication in PLOS ONE. Congratulations! Your manuscript is now with our production department. 

With kind regards,

on behalf of

Professor Constantinos I. Siettos 

Academic Editor

PLOS ONE